# A Community of Practice Approach to Teaching International Entrepreneurship

**Martina Musteen [1], Ross Curran [2], Nuno Arroteia [3], Maria Ripollés [4] and Andreu Blesa [4,\*]**

[1] Fowler College of Business, San Diego State University, San Diego, CA 92182-8346, USA;
mmusteen@mail.sdsu.edu

[2] School of Social Sciences, Heriot-Watt University, Dubai Campus, Dubai International Academic City,
Dubai P.O. Box 294345, UAE; ross.curran@hw.ac.uk

[3] Coventry Business School, School of Strategy and Leadership, Priory Street, Coventry CV1 5FB, UK;
ac9506@coventry.ac.uk

[4] Department of Business Administration and Marketing, Universitat Jaume I, 12071 Castelló de la Plana,
Spain; maria.ripolles@uji.es

\* Correspondence: blesa@uji.es; Tel.: +349-6438-7118

**Abstract:** With a dearth of research on international entrepreneurship pedagogy, there is a gap in knowledge on the effectiveness of educational programs, courses, and teaching methods in stimulating and promoting international entrepreneurship practice. To address the gap, this study evaluates an experiential teaching innovation in the area of international entrepreneurship, the Global Board Game project. Designed as a Community of Practice (CoP), the project provides students the opportunity to participate in the construction of their knowledge through interactions with their counterparts in other countries. A qualitative analysis of student essays indicates that the Global Board Game project is effective in helping students achieve learning outcomes, which include defining, recognizing, and evaluating international business opportunities; designing and validating a business model based on such opportunities; and creating a plan for pursuing these opportunities. Additionally, it indicates that participation in the project enhanced students' attitudes toward entrepreneurship as a career path.

**Keywords:** international entrepreneurship education; Global Board Game project; entrepreneurial intention; active learning; Community of Practice; international student teams

## 1. Introduction

Research in international entrepreneurship (IE) has grown considerably during the last decade, primarily investigating the antecedents and outcomes of early internationalization (Jones et al. 2011; Keupp and Gassmann 2009; Zahra and George 2002). Interestingly, little research has been devoted to IE pedagogy, leaving a significant gap in knowledge related to the link between the courses and teaching methods on educational programs, and IE practice. This article aims to address this gap by evaluating an experiential teaching innovation in the area of IE—the Global Board Game Project (GBGP). Specifically, given the dearth of theory regarding IE pedagogy, we use the partially grounded approach to assess this teaching innovation, designed as a Community of Practice (CoP), through analysis of students' self-perception of their abilities related to defining, recognizing, and evaluating international business opportunities; designing and validating a business model based on such opportunities; and creating a plan for pursuing these opportunities. Our study also provides some evidence that, by promoting learning through practice, the CoP-based teaching method has impacted upon students' emotions, self-efficacy, and self-perceptions of their entrepreneurial intentions.

As we describe in greater detail in the next section, the design of the GBGP was rooted in IE literature. The GBGP involves semi-structured online collaboration between undergraduate student teams from three different countries to ideate, develop, and market a product (a board game) to another country. The emphasis is on communication within and between teams to learn, experiment, and test assumptions towards creating a tangible board game prototype and developing a viable market entry plan. The IE literature informed the intended outcomes of the teaching method in terms of specific knowledge domain, students' attitudes, and mode of instruction. We therefore followed the social constructivist theory of learning (Bandura and Walters 1977), which suggests effective student learning requires opportunities for them to develop an active role in the construction of their knowledge through interactions with others (Bae et al. 2014; Nabi et al. 2017). This perspective is in contrast with previous thinking, in which knowledge was thought to be learned in a classroom and then seamlessly transferred to a real-life setting. To this end, the GBGP was designed to allow students to experiment (and learn) in the process of ideating and creating a real product (a board game) with the aim of selling it in a foreign market. Involving students from three different universities in three countries, the GBGP was designed around a CoP, which can be defined as a collaborative approach to learning, and through practice, facilitating both knowledge sharing and creation within a specific domain. By adopting the CoP approach, we sought to respond to calls from the literature (Fayolle 2013; Wiklund et al. 2011) that entrepreneurial education research and practice should be more theory-driven. By examining the impact of the CoP-based teaching innovation on IE-related learning outcomes and on entrepreneurial intention, as well as discussing the challenges of the process, we seek to, at least partially, address this gap in the IE literature.

We offer evidence showing students found the CoP format adopted in the GBGP effective by providing them with the specific skills applicable in the area of international venturing. Furthermore, our findings suggest that a number of participating students attributed their greater desire to become an entrepreneur to participating in the GBGP. This contributes to both the IE literature and the literature on CoP (Zhang and Watts 2008), which has suggested that CoPs can be used effectively to foster learning in an online environment.

The paper is structured as follows. First, we review the relevant literature on IE that informed the design of the GBGP. Second, we discuss the rationale for utilizing the CoP framework as an approach to promote entrepreneurial learning in an international context. This is followed by a brief description of the design and implementation of the GBGP and the method used to evaluate its efficacy as an IE teaching innovation. We conclude by presenting findings of the analysis and discuss their implications for international entrepreneurial learning.

## 2. International Entrepreneurship in Higher Education

In reviewing the literature relevant to the IE pedagogy that informed the design of the GBGP, we identified three core elements for designing a teaching tool that would be relevant and effective for students of IE. These include the content domain, student attitudinal outcomes, and the mode of delivery.

*Content Domain.* The meaning of the term "international entrepreneurship" has evolved over the past few decades and the research on the topic has grown substantially, primarily focusing on international new ventures (INVs) and "born globals" (Andersson 2011; Jones et al. 2011; Schwens et al. 2018). In their seminal article, Oviatt and McDougall (1994) defined an INV as "a business organization that, from inception, seeks to derive significant competitive advantage from the use of resources from and the sale of outputs to multiple countries" (p. 49). Using this definition, the subsequent IE literature has investigated the antecedents of new ventures' early internationalization. In light of criticism that this IE definition excludes corporate entrepreneurship in international markets (Zahra and George 2002), McDougall and Oviatt (2000) later proposed a new definition in which IE was conceptualized as innovative, proactive, and risk-seeking behavior (Covin and Slevin 1989) that crosses national boundaries. In addition, it was suggested that a critical element of IE is the development of an entrepreneurial orientation and value creation. Incorporating more elements from the core entrepreneurship field, the subsequent IE literature continued to focus more on the concept

of entrepreneurial opportunities as the main defining element of IE. Specifically, IE scholars pointed to Shane and Venkataraman's (2000) definition of the study of entrepreneurship as "the examination of how, by whom, and with what effects opportunities to create future goods and services are discovered, evaluated, and exploited". Accordingly, Oviatt and McDougall (2005, p. 540) modified their IE definition further by proposing that "international entrepreneurship is the discovery, enactment, evaluation, and exploitation of opportunities—across national borders—to create future goods and services". This definition of IE has been widely accepted by scholars (Jones et al. 2011; Schwens et al. 2018), and thus was also used as a basis for the GBGP.

Given the emphasis on market validation in the entrepreneurship literature (Shane 2003; Martin et al. 2013), we also incorporated foreign market validation as an important element in the GBGP and included planning and implementation of foreign market entry strategies (Unger et al. 2011). With that in mind, the GBGP was designed to provide students with knowledge in three specific areas: (1) defining, recognizing, and evaluating international business opportunities; (2) designing and validating a business model for such an opportunity; and (3) creating an offering for and translating the proposed business model to a specific international market.

*Students' Attitudes.* Besides providing students with appropriate content in specific knowledge domains, the aim of entrepreneurship education to increase students' intention to own or start a business is another important outcome of entrepreneurship education (Bae et al. 2014; Souitaris et al. 2007). According to the theory of planned behavior (Ajzen 1991; Ajzen and Fishbein 1977), as students enroll in entrepreneurial education, they should be exposed to examples of successful business planning and proactive interaction with successful practitioners. These pedagogical elements facilitate coping strategies, which help maintain motivation and interest, leading to greater expectations of success and increased entrepreneurial self-efficacy (Noel 2002). Indeed, the meta-analysis developed by Martin et al. (2013) supported the positive relation between entrepreneurship education and entrepreneurial intention. Thus, increasing students' intentions to become international entrepreneurs was also a desired outcome for the GBGP.

*Mode of Delivery.* As a pedagogical method, the GBGP was designed to emphasize situated learning on the assumption that learning should be driven by students (Bandura and Walters 1977), that much of what is learned is specific to the situation in which it is learned (Lave and Wenger 1998), and that interactions with others are an important learning element (Béchard et al. 2005). Accordingly, we followed the precept that teaching should be conceived as a strategic intervention to create learning environments, facilitating practice and knowledge transfer among students (Neck et al. 2014). This way of understanding teaching in entrepreneurship gives teachers a role that is different from the traditional role of transmitting knowledge. Teachers must become the manager and facilitator of the student learning process (Löbler 2006). Accordingly, the GBGP was designed to enable students to develop situated learning in the process of ideating and developing a product (a board game) and proposing a business model used to sell it in a foreign market.

## 2.1 Defining Community of Practice

In order to facilitate students' situated learning, we adopted the Community of Practice (CoP) approach for the GBGP with the broad aim to promote creation of knowledge derived from methods of learning through practice (Handley et al. 2006). A CoP can be defined as a group of individuals—the community—who share their interests on a specific topic—the domain—and gain a greater degree of knowledge and expertise on that topic through regular joint experimentation—the practice (Wenger 2011). Through the practice, the practitioners share information and develop knowledge resulting from the members' engagement in joint practical activities and discussions (Wenger 1998).

There exists an extensive theoretical and practitioner-oriented literature advocating CoP as a collaborative approach for promoting situated learning (Wenger 2000) within educational and other organizational contexts (Arthur 2016; Ceptureanu and Ceptureanu 2015; Harris et al. 2017; Howlett et al. 2016; Koliba and Gajda 2009; Pharo et al. 2014; Tight 2004). Scholarship in CoP is mainly based on Wenger's initial research (Wenger 1998; Wenger and Snyder 2000; Lave and Wenger 1998), but also on his latest conceptualization (Wenger 2000, 2011). Initially, CoP has been thought to be based

on self-selection with members informally bound together by their interest in undertaking joint learning (Lave and Wenger 1998; Handley et al. 2006), named here as emergent CoP. However, later Wenger (2011) suggested an instrumental usage of CoP in practice reflecting his transition from an analyst of social learning systems to a designer of them (Clegg 2012). Instead of an individual's identity becoming aligned with his or her CoP, the CoP becomes a way of connecting individual identities to the achievement of collective learning aims (Arthur 2016). In spite of being organic and self-directed, deliberate CoPs can be defined and cultivated explicitly by organizations to achieve specific learning goals. Thus, CoPs can also be applied as a specific methodology used for particular learning purposes, specifically when the joint practice is supposed to be an important learning element, engendering situated learning (Handley et al. 2006; Pharo et al. 2014). Examples of this instrumental usage of CoP can be found in organizations to foster knowledge sharing among employees, named as organizational communities of practice (Ceptureanu and Ceptureanu 2015; Koliba and Gajda 2009; Lee and Williams 2007; Aljuwaiber 2016). Authors such as Hodge et al. (2014), Howlett et al. (2016), and Tight (2015) support the usage of deliberate CoP in an educational context to enhance students' situated learning. In summary, the deliberate CoP approach can be considered in an educational context to offer a space of learning in which students can experiment with different tasks requiring communication and interaction among them. Table 1 below summarizes the key differences between emergent and deliberate CoP.

**Table 1.** Differences between emergent Community of Practice (CoP) and deliberate CoP in an educational context.

| Element | Emergent CoP | Deliberate CoP |
|---------|--------------|----------------|
| Task mission | Emergent from the community. | Assigned by the instructor. |
| Membership | Voluntary and dynamic. | Appointed. Defined by the instructor. |
| Participation | Variations of degree of participation are permitted. | Full participation is recommended but different degrees of commitment are permitted. Participants are allowed to developed different team roles. |
| Activities | Coming from members. | Provided by educational institutions |
| Structure | Emergent. | Emergent. |
| Resources | Coming from members. | Provided by educational institutions and coming from members according to its degree of engagement |

Source: Adapted from Aljuwaiber (2016) and Wenger (2011).

In deliberate CoPs, teachers act as catalysts for students' situated learning to emerge (Viskovic 2006). Accordingly, they are responsible for the students' mutual engagement; for promoting the sense of joint enterprise, for developing the repertoire of activities, and for providing infrastructure that will support such communities and enable them to fulfill their learning objectives (Wenger 1998; Wenger and Snyder 2000). Additionally, teachers are likely to use non-traditional methods to assess the value of the CoP (Wenger and Snyder 2000).

Thus, deliberate CoP requires a physical or virtual space that facilitates interaction among CoP members (Koliba and Gajda 2009). Spaces can be created through a formal or informal designation of physical meeting times and places, or virtually, as space for ongoing dialogue without being mediated by a third party. This space forms the basis through which a "shared repertoire" for the group emerges. CoPs with international participants (like the GBGP) generally demand virtual spaces. Given that the GBGP relies primarily on new information and communication technologies and internet capabilities, it can be considered a virtual CoP (Dubé et al. 2005).

While online environments present both challenges and opportunities for CoP development, there is evidence that the limitations inherently associated with online environments and CoP development can be overcome, enabling them to also become effective settings for CoP (Zhang and Watts 2008; Koliba and Gajda 2009). The next section details how the GBGP incorporated these elements and describes the overall CoP design.

## 3. Methodology

### 3.1 The Global Board Game Project as the Community of Practice

The GBGP involved students in undergraduate international entrepreneurship or marketing classes. Student teams of four to five students were formed in each of the participating universities—*San Diego State University* (SDSU) in the USA, *Abertay University* (AU) in the United Kingdom, and *Universitat Jaume I* (UJI) in Spain. There were 22 teams in total—10 teams in SDSU, 6 in AU, and 6 in UJI. These teams were then paired with a foreign partner team (FPT) from an overseas university (five U.S. teams were paired with five U.K. and five Spanish FPTs, and there was one United Kingdom–Spain FPT dyad). Each participating team was tasked with identifying an opportunity for a board game product, developing a prototype of that product that is calibrated for their partner team's domestic market, where the FPT served as a local agent offering unique insight into the specificities of their context. Through this approach, each team fulfilled both entrepreneurial product development as well as foreign partner/agent roles for the semester.

During the course of the project, following the CoP principles, student teams entered into a dialogue with their overseas partners and were encouraged to use their FPT's understanding of the local market to inform the design process of a newly developed board game product. To fully realize the usefulness of FPT communication, students engaged in initial desk research to gain an understanding of the political, economic, social, and technological factors affecting their allocated target market with the purpose of identifying a viable business opportunity. This research process and the accompanying product development element of the project were supported through the delivery of a series of globally aligned tasks/worksheets completed by all participant teams. The developmental worksheets were aligned with the three IE content domains and designed to stimulate FPT engagement, and to capture the information gained through the FPT communication process. The worksheets served to scaffold the development of an informed market entry plan for a newly developed board game product and were the micro-foundations for the development of the students' entrepreneurial competencies. Table 2 below summarizes the timeline followed as part of the GBGP.

**Table 2.** Global Board Game Project (GBGP) timeline.

| Timeline Day/Month | Action Point | Globally Aligned Student Tasks |
|---|---|---|
| 18/09 | GBGP team formation. | Foreign partner team (FPT) dyads created. |
| 18/09 | Ideator.com platform profiles created. | Ideator.com profiles created and synced with FPTs. |
| 25/09 | Early viability check of board game concepts. | Critical reflection on game concepts. |
| 06/10 | Foreign market assessment | Research undertaken into relevant markets, in FPT locations—Worksheet I |
| 13/10 | Ideation and creation of prototypes/mockups of the board game | Development of physical version of the prototype and/or submit order for production to send to FPT. Game instructions communicated to FPT—Worksheet II |
| 19/10 | Competitive positioning | Worksheet I and II completed and shared. |
| 20/10 | Competitive positioning | Presentation of Worksheet I and II. |
| 27/10 | Business model canvas Lean market entry action plan | Worksheet III and IV. |
| 30/10 | Lean market entry action plan | Deliver Worksheet IV to FPT. |
| 03/11 | Product test and feedback to FPT | Worksheets V and VI. |
| 10/11 | Feedback to FPT | Send and receive feedback from partner team on product and Lean Market Entry Action Plan—Worksheet VI |
| 29/11 | Peer assessment | Peer assessment—Worksheet VII |
| 14/12 | Final presentation | Final presentation Global Board Game Project to lecturers and colleagues |
| 21/12 | Reflective summary | Submission of short personal reflective essay. |

The collaboration within and among student teams was further encouraged and enabled through the use of an online platform (Ideator.com). *Ideator.com* facilitated both intra- and inter-team communication, and gave instructors the ability to oversee and monitor the level of student communication activity. The *Ideator.com* platform also provided a virtual space for the creation of team venture profiles akin to those developed by entrepreneurs seeking to bring product concepts to market through support and collaboration with other entrepreneurs. An additional advantage of the platform was that it allowed students to upload and record their completed worksheets in a transparent manner, enabling other members of the cultivated online community to learn from one another's outputs.

Table 3 below summarizes alignment of the GBGP design with the CoP elements along with the supporting literature, highlighting the delivery team's efforts to cultivate and facilitate an organic CoP among GBGP participants.

**Table 3.** Implementation of a Community of Practice model in the Global Board Game Project. SDSU—San Diego State University (SDSU); AU—Abertay University; UJI—Universitat Jaume I.

| Community of Practice Dimension | Definition | Manifestation in Global Board Game Project | Supporting CoP Literature |
|---|---|---|---|
| **Domain** | Dynamic, actively evolving context in which shared interests emerge. | Students enrolled in international entrepreneurship and marketing courses at SDSU, UJI, and AU. | Wenger (1998, 2000); Wenger and Snyder (2000). |
| **Community** | Mutual member engagement in a shared interest. Community identity is generated through shared efforts imbuing shared identities. | A CoP was purposely cultivated through stimulation of intra/inter team interaction and collaboration to meet the goals of the project. | Arthur (2016); Pharo et al. (2014); Wenger (2011); Wenger and Snyder (2000). |
| **Practice** | Outputs of community-member engagement in the CoP towards a common goal. | Students completed a worksheet/task series and developed a board game product calibrated for their FPT's market. | Ardichvili et al. (2006); Wenger (1998); Wenger and Snyder (2000). |
| **Space** | Infrastructure required to facilitate CoP activity, e.g., online collaboration tools as well as the time required to engage in a CoP. | Student teams utilized the *ideator.com* website as an online collaboration tool to facilitate intra/inter team working and CoP activity. Student teams were allocated time during the semester to participate in the CoP. | Koliba and Gajda (2009); Zhang and Watts (2008). |

### 3.2 Data Collection and Analysis

To assess the degree to which the GBGP performed as an effective CoP in terms of student learning outcomes and impact on entrepreneurial intention, we analyzed students' essays (up to 500 words in length) submitted at the end of the semester. The essays included students' personal reflections on their experiences developing an idea, designing a product, and developing a marketing plan to sell the board game in a foreign country. Students were instructed to focus on the learning aspects of the GBGP and how their group overcame project related challenges. They were also invited to share how they felt during the process. All texts were in English, which was the language used by the GBGP participants. It is important to note that in order to motivate the students to submit their summaries, they accounted for a small percentage (around 2%) of the final grade. At the same time, to minimize social desirability bias, we assured the students that they could reflect on both positive and negative aspects of the project and would not be penalized for any negative comments offered. In addition, we emphasized that their insights will be used to improve the project. We collected and examined all reflective summaries that were submitted by the students—86 in total (30 from SDSU,

35 from UJI, and 21 from AU). Twenty-three students did not provide any summaries and were thus not considered in the study.

Given that the theory related to IE pedagogy is still underdeveloped, we adopted a qualitative, "partially grounded", inductive approach to our analysis (Jack et al. 2008; Sunduramurthy et al. 2016). The partially grounded approach differs from the purely grounded approach to data analysis in several respects (Ahsan et al. 2018; Shepherd and Sutcliffe 2011). The purely grounded approach is used to derive new theory solely from data (Glaser and Strauss 2017). Researchers pursuing the purely grounded, inductive approach often take a position of "unknowing" (Shepherd and Sutcliffe 2011, p. 361), approaching the raw data without preconceived notions about the meaning they are searching for. The partially grounded approach is also inductive in that data is used to better understand a phenomenon and derive concepts and constructs. However, the difference is that the analysis is guided by existing literature. In other words, the partially grounded approach connects and compares the meaning emerging from the raw data with existing concepts and constructs (Ahsan et al. 2018). Specifically, the partially grounded approach begins with a research question and some theoretical a priori understanding of the phenomenon; however, it is flexible enough to "let the data speak", allowing for new insights to emerge (Shepherd and Sutcliffe 2011). Using this approach, we drew on our review of the IE and CoP literature regarding the general domains of the student learning in IE, as summarized in the previous section. Specifically, these included students' acquisition of practical insights related to (1) defining, recognizing, and evaluating international business opportunities; (2) designing and validating a business model for such an opportunity; and (3) creating an offering for and translating the proposed business model to a specific international market. Consistent with our aim to leverage the CoP design, we looked for evidence of learning through knowledge sharing and practice. Lastly, informed by the previous literature on entrepreneurship education (Baidi and Suyatno 2018; Nabi et al. 2017; Susetyo and Yuliari 2018), we also sought evidence of the impact of GBGP participation on students' self-perceptions of their entrepreneurial intention. In addition to these themes derived from previous literature a priori, the research team was open to additional themes emerging from the analysis.

Eighty-six reflective summaries were collected and analyzed qualitatively in several phases (Miles and Huberman 1994). In the first phase, following Jack et al. (2008), the members of the research team independently read and re-read the text while taking notes on the theoretically pre-specified constructs/domains as well as new, emergent themes. Second, with the aim of gaining an initial holistic understanding of the student experience during the GBGP, initial impressions and notes were discussed among the research team members. In the third phase (open coding), three researchers independently coded the textual data, categorizing specific quotes into 28 different categories maintained in RQDA, a software package that is widely used by qualitative researchers. These concepts were either in vivo, occurring frequently in the analyzed texts, or were given labels based on the researchers' interpretations of emerging concepts (Sundaramurthy et al. 2013). These were then revisited by the team along with the reflective summary text. Only quotes attributed directly to participating in the GBGP were retained. Quotes related to learning not due to attending the course were eliminated. In the last phase (the axial coding), the research reviewed the coding in order to resolve inconsistencies and identify a consensus in the identification of learning results directly related to participation in the GBGP. It also involved the research team in revising these codes by comparing, discussing, and debating the themes and concepts that emerged during independent analysis. This process also involved giving meaning to data that were fractured during the open coding phase (Strauss and Corbin 1988).

This process resulted in retaining 567 quotes affiliated with 9 broad categories, some of which had more than one dimension. Table 4 presents the summary of the categories and their dimensions, illustrative quotes, and the total number of quotes.

**Table 4.** Results of the qualitative analysis.

| Constructs | Dimensions | Illustrative Quotes | Number of Quotes |
|---|---|---|---|
| Defining, recognizing, and evaluating international business opportunities | Opportunity identification | "[…] I feel like we came up with a new idea that is unlike anything in the market." (SDSU) <br> "First, we asked ourselves "who are the customers?", this gave us the idea of designing a drinking board game that would appeal to university students aged 18–25 as we felt like we were familiar with the market, being university students ourselves…" (AU) <br> "I discovered how important is to very well understand our target market's needs and problems to accurately and interestingly present our product and find the best way of problem solving." (UJI) <br> "[…] we came up with the idea of a fun, interactive drinking game quite quickly." (AU) | 29 |
| | Opportunity validation | "Definitely the fact that it is necessary to adjust and adapt for a business to become successful. An initial idea might not be the best one, so testing and learning plays a role in the success of a business." (SDSU) <br> "It forced us to research and think outside the box while considering what the public would be interested in." (AU) <br> "After understanding it and collecting knowledge on the product, we needed to work out a proper marketing plan figuring out indicators of our target market and the competitor's situation." (UJI) | 22 |
| Designing and validating a business model for such an opportunity | Business model design | "I am proud to say that I am able to write the business model canvas and develop it by interacting our FPT, asking them some international cultural business issues such as best distribution channel, easiest way for customer relationship and how to engage with this plan to achieve the objectives." (UJI) <br> "Now, I feel like I can successfully create a business plan by knowing exactly what my product is, who the target market is, how to find them and how to come up with a general idea for sales and revenue." (SDSU) | 24 |
| | Business model validation | "All these hypotheses were contrasted, and corrected if necessary, thanks to interviews we conducted with people who fulfilled the requirement of our supposed target audience." (UJI) <br> "We initially thought selling our games at gyms would be our target market but after speaking to our foreign partners we learned that people in Spain would most likely not purchase this game at their gym because there is not many gyms and they don't sell any merchandise at the local gym." (SDSU) <br> "[…] it is necessary to adjust and adapt for a business to become successful. An initial idea might not be the best one, so testing and learning plays a role in the success of a business." (SDSU) | 25 |
| Creating an offering for a specific international market | | "I understood that in order to deal with an international market it is fundamental to have a product which can be adjusted in relation to the cultural needs of the country taken into account." (SDSU) <br> "I discovered how important is to very well understand our target market's needs and problems to accurately and interestingly present our product and find the best way of problem solving." (UJI) <br> "One of the biggest challenges for us, was that after we decided to create a drinking game, looking at the legal issues we had to face. We understood that the drinking age in America was 21." (AU) | 35 |
| Translating the proposed | | "The partners were extremely helpful in guiding a marketing plan in Spain. […] They provided specific | 42 |

| | | | |
|---|---|---|---|
| business model to a specific international market | | events, and suggested we could modify the message to fit each event." (SDSU)<br><br>"In America this would make sense, however the feedback that we got suggested that we take the game and distribute it to the sports teams because they act like fraternities in America." (SDSU)<br><br>"The idea to develop a plan of marketing, to choose the best international strategies, to decide whether to create alliances with other enterprises, allow us to have a complete vision of as the reality it works to today." (UJI)<br><br>"Overall I feel like the course encouraged me to do a lot of independent learning. Particularly in non-academic ways, such as services provided by third party companies such as Amazon and their fulfilment service, and how to best use crowdfunding services and other modern distribution methods." (AU)<br><br>"Developing a marketing plan was a bit of a challenge since we didn't have any knowledge about marketing channels in Spain. That was also one the areas where we got the most help from our Spanish partner team." (SDSU) | |
| Situated Learning | Knowledge Sharing | "They [the partner team] also contributed new ideas giving us their opinion helping us in this way to improve our work." (UJI)<br><br>"After this scenario, I am now more knowledgeable about just how important communication is within a business, especially if the person or group being worked with isn't within a relatively close distance." (AU)<br><br>"However, throughout the course, we found better ways of working with each other and were able to improve our skills of intercultural communication." (SDSU)<br><br>"The creative conversation helped us with our team interactions and provided helpful insight into the different ways we could create the product." (SDSU)<br><br>"Participating in the global game project was an interesting, exciting and useful experience because it gave me the opportunity to grow personally, gaining new knowledge, skills and competences and collaborating with a group of people, sharing tasks, comparing ideas, exchange any feedback on the respective works and then share the same goals." (UJI)<br><br>"It was a great experience working together so close with people you have not known before and getting to know them, building up trust and work spirit to form a productive team. When it comes to working with the foreign partner team, we also only made positive experiences as they always helped us answering all our questions." (UJI) | 122 |
| | Learning through practice | "The mix between theory and practical approach during the board game project was a good addition and had a quiet good learning effect. For me it made totally sense to work during our "marketing plan" on a real example and not just in a theoretical example." (AU)<br><br>"Applying the theory to practice I have learned how to do business, how to create a company, from the idea and production of the product to bring it into the market (where to sell it, how to distribute it, at what price, etc...)." (UJI)<br><br>"Furthermore, it was a very learning exercise to actually develop a product and bring it from the drawing board into a sellable outcome in terms of a final product." (SDSU) | 54 |

| | | |
|---|---|---|
| | "Especially the fact that we were able to apply our acquired knowledge to a real product development and shipping was great and differentiated this class from other classes that focus only on theory or on projects that are not really executed." (SDSU)<br>"The Global Group Project also taught me a great deal about the theoretical issues such marketing, sales, international cultural business issues etc." (SDSU)<br>"One thing that I really liked was the practical part when it came to testing and evaluating the foreign partner team's board game." (UJI) | |
| Students' intention to become international entrepreneurs | "I look back and see everything I have developed and achieved in this subject, also creating more desire to enter the real world and create my own own product in the future through the process that I have been doing during the semester." (UJI)<br>"My team and I enjoyed the course so much, we plan to continue working on our game, using what we have learnt as a base to grow on. We hope to possibly launch our game in 2018/19." (AU)<br>"This opportunity is very relatable to a career I could see myself pursuing therefore giving me valuable experiences." (AU)<br>"It was very rewarding to finalize our strategy and successfully hit our goals each week with them. I am excited to take some of what I learned here and implement it in the startup I work at." (SDSU) | 33 |
| Self-efficacy | "I feel considerably more prepared to face the design of an idea in the future." (UJI)<br>"Although selling to another country may seem difficult I feel a lot more confident to be able to do this one due to this project, I am grateful." (SDSU)<br>"As a result of the GBG project, I feel more confident with my understanding in working with foreign markets than I did walking in to this class." (SDSU) | 35 |
| Emotions | "At first when I saw what we have to do I felt a little scared for if I do not know how to do it, but finally I think that in general it has gone well." (UJI)<br>"From the moment I understood the content of the module, I couldn't wait to get started. I thought the idea of creating your own product, marketing it, and physically selling it, was very exciting." (AU)<br>"It was exciting to be able to express our entrepreneurial traits by having the flexibility to create anything we want and to employ our creativity skills in to a project like a real business project being launched by a group of entrepreneurs." (SDSU) | 27 |
| Cross-cultural Competences | "That improved my cultural understanding and taught me how to interact with these cultures." (UJI)<br>"This has helped me broaden my perspectives about etiquettes in different nations." (SDSU)<br>"I feel that having reached the end of this module I have gained valuable experience and knowledge of the business world and how to be creative." (AU)<br>"It not only enhanced my entrepreneurial mindset, but also introduced me to dealing with foreign markets." (SDSU)<br>"I have also learned that you have to be flexible and patience when it comes to international businesses and entrepreneurship. My cultural knowledge has increased during this project." (SDSU) | 92 |

| | | |
|---|---|---|
| | "Another key point of the project was the experience of working with people from outside which made us integrate into their culture and investigate another market different from the one we used to use." (UJI) | |
| Challenges | "We have learnt that communication through time zones can be quite challenging and that personal interaction in many cases would be a better solution then email." (SDSU)<br>"To ensure frequent communication with my partner team, I extended my university day by an hour or two." (AU)<br>"It has also been difficult for me to communicate with our US companions, by the difference in schedule and the delay in answering via email." (UJI)<br>"[…] our communication with the destination country, United Kingdom, with the students of the Abertay University, has not been good." (UJI)<br>"During our group work we had the typical problem; not every team member showed the same engagement while developing the game and preparing the presentation." (AU)<br>"Since our whole team was enthusiastic about developing a real board game, we managed to make the most of our meetings." (SDSU)<br>"Having worked in a group has certainly brought many advantages, because I do not think I would have been able to do everything myself, but from this point of view I expected more collaboration, especially when me and my classmate, we needed advice or clarify certain things, then I would have preferred more communication." (SDSU)<br>"The idea of setting up a business and making an own board game is really interesting but the collaboration with the other university went quite difficult." (UJI) | 119 |

### 4. Study Findings

The partially grounded approach to evaluation of the GBGP as an IE pedagogical tool provided some interesting insights. First, as Table 4 shows, consistent with the intended design aimed at the key domains of the IE field, the textual analysis of student reflection summaries indicates that students participating in the GBGP felt they engaged in processes related to the definition, recognition, and evaluation of international business opportunities (a total of 51 quotes). Based on the number of quotes, it appears that the GBGP was effective in stimulating students' efforts in identifying international opportunities (29 quotes) and, to a somewhat lesser degree, validating such opportunities (22 quotes). Specifically, students expressed sentiments such as "[we] *came up with a new idea that is unlike anything in the market*" and "[we] *developed an idea that seemed fun and interesting*". In other words, most of the GBGP participants felt the project provided them with space to search for and define a business opportunity in the board game market (e.g., "*we first had to think about a market and what goal we wanted to reach*"), but they also committed themselves to gathering additional data to "*adapt and modify [the product] in the early stages to create the best idea possible*".

Drawing on the IE literature, we also sought evidence for students' learning to design and validate a business model based on their idea. Forty-nine quotes were identified to be related to this theme. Twenty-four related to business model design, while 25 quotes reflected students engaging in business model validation. For example, students felt that through GBGP participation, they acquired knowledge that made them feel "*able to write the business model canvas and develop it by interacting with* [the partner team]". They also referred to the need to identify the principle components of a business model by understanding "*what my product is, who the target market is, how to find them and how to come up with a general idea for sales and revenue*".

Our analysis of the reflective summaries also suggests that the GBGP gave students the opportunity to practice creating an offering for a specific international market (35 quotes), which required taking into account the specific characteristics and differences of that market. For example, one student commented, "*Throughout the whole process I also experienced that when you design a product you have to consider the impact of culture*". Another stated, "*I have learned that you can't just design a product but have to adapt it to the target market in order to make it appealing*".

Besides creating an offering (i.e., a product) aimed at international markets, participation in the GBGP also required the key components of a business model to be adapted so as to suit the specific international market. The textual analysis returned 42 quotes related to this theme, indicating that students had to take into account how to market and distribute their product and use partnerships and alliances to enter the market.

Consistent with the CoP design, a large number of quotes (176) were identified to indicate the GBGP was effective in stimulating situated learning through knowledge sharing and learning through practice. Specifically, in 122 quotes, the students expressed sentiment that their interaction with their counterparts in another country was valuable in the process of gaining new knowledge and skills related to IE. For examples, the quotes suggested that the "*creative conversation*", "*working together*" across time and space, and "*building up trust*" among student teams contributed to the acquisition of *"helpful insight"* and "[improving] *skills and competences*". Moreover, an additional 54 quotes provided preliminary evidence that the CoP approach to learning via the GBGP project enabled students to learn through practice. "*Applying theory to practice*" and engaging in "*a different way to learn and to be able to practice all the theory learned previously*" represent some examples of student sentiment relating to the GPGP as a tool for learning about IE.

Informed by the previous literature on entrepreneurship education, the partially grounded analysis also sought to discover evidence of students' self-assessment of their attitudes toward becoming international entrepreneurs and/or pursuing IE in the future. Thirty-three quotes were identified to be consistent with this theme. Specifically, students expressed their desire either to pursue the project after the semester end (e.g., "*launch our game in 2018–2019*") or to "*create my own product in the future*".

In addition to the a priori themes discussed above, there were several other themes that emerged from the textual analysis of the student reflection summaries. First, 92 quotes indicated that students appreciate the opportunity that GBGP provided in terms of their cross-cultural competencies. Some of the comments, such as "[the project] *enhanced my understanding of differences between cultures*" and "for *me especially the cultural aspects of these projects are interesting. I love the differences in culture and this project took that completely into account*", indicate that GBGP participation made students aware of the impact of cross-cultural differences affecting the creation of new products for foreign markets, and accompanying sales and marketing plans. Second, 35 quotes related to student's self-efficacy. Specifically, students expressed confidence that participation in the GBGP enhanced their ability to deal with the challenges of IE such as product design or working in a cross-cultural setting. This finding is not entirely surprising given the strong theoretical link between entrepreneurial self-efficacy and entrepreneurial intention (Baidi and Suyatno 2018; Noel 2002; Susetyo and Yuliari 2018). Our preliminary findings suggest that the self-efficacy is also important in the IE domain. Third, 27 quotes have indicated that students participating in GBGP felt a variety of emotions. For example, students felt "*scared*" and "*excited*" at the beginning of the project, experienced "*ups and downs*", "*moments of stress*", and being "*disoriented*" during the various stages of the project. Nevertheless, students reported they found the overall experience rewarding. Lastly, the analysis resulted in identification of 119 quotes related to various challenges experienced by GBGP students. In particular, they highlighted difficulties communicating across different time zones and not always getting a timely response from their counterparts in the other country.

Overall, students' comments suggest the GBGP has been quite effective in terms of enhancing entrepreneurial knowledge for international venturing. Given its design as an online CoP, students from different universities in three countries were able to interact purposely and, in the course of this interaction, gained knowledge on IE. As the students' comments reveal, the CoP approach was successful in fostering situated learning through practice and information sharing. In fact, the analysis of the student reflections indicates online CoPs are not limited to tacit knowledge sharing, but also include explicit knowledge (Lave and Wenger 1991). Consistent with Aljuwaiber (2016), we found evidence that the online CoPs (in our case, utilizing the *Ideator.com* virtual space) can contribute to sharing of both tacit and explicit knowledge in the IE domain. However, that is not to say that this process was without challenges.

## 5. Discussion and Conclusions

The purpose of this article was to address a gap in the IE literature by reporting on the evaluation of a novel teaching tool, the GBGP. Involving student teams from three institutions in three countries, the GBGP was designed to provide students with practical knowledge in three key areas: (1) defining, recognizing, and evaluating international business opportunities; (2) designing and validating a business model for such an opportunity; and (3) creating an offering for and translating the proposed business model to a specific international market. Adopting the CoP approach (Wenger 2000), the GBGP involved creating an online space and was structured to facilitate experimentation and knowledge exchange among participating students (Neck et al. 2014).

The qualitative analysis of student reflections on the project provided preliminary evidence that the GBGP, as a teaching tool, was effective in helping students create a more tangible link between IE theory and practice. In addition, it indicated that participating in the project influenced students' attitude toward entrepreneurship as a career path. Specifically, a number of them expressed the desire to start their own business or pursue the project further after the end of the semester. However, these findings must be viewed and interpreted with caution. While we find evidence that a number of students have expressed that participation in the GBGP made them consider or even desire pursuing entrepreneurial careers, we cannot claim that these students will indeed become entrepreneurs or that their intention will persist. This is a weakness that is typical of many studies on entrepreneurial education (Nabi et al. 2017).

Besides some preliminary evidence that CoP design of the GBGP had impact on student learning in the area of IE and stimulated situated learning through practice and collective information

exchange, our findings provide some additional insights related to some unexpected, emergent themes. These include the GBGP impact on student cross-cultural learning, self-efficacy, and the emotional experiences accompanying participation in the project. While investigated in the context of pedagogical interventions (Tuleja 2014), the role of cross-cultural competencies has been relatively understudied in the area of IE in general, and IE pedagogy in particular. Our preliminary findings suggest that working in the context of a virtual, cross-cultural CoP such as GBGP may enhance students' cross-cultural intelligence. Both self-efficacy and emotions have been studied in the general entrepreneurship literature (Baidi and Suyatno 2018; Baron 2008; Biraglia and Kadile 2017; Noel 2002); however, neither has been thoroughly examined in the context of IE pedagogy. While very preliminary, our study suggests that to simulate the IE process in the classroom effectively, students need to be exposed to tasks that, to some degree, will engender both negative and positive feelings but eventually enhance students' confidence in themselves to complete such tasks.

The analysis also pointed out some challenges—mainly related to communication problems between the partner teams. This leaves a room for ongoing improvement and modification of the project. Specifically, the GBGP participants may need to be better prepared to deal with communication difficulties related to different time zones.

Our study suggests that GBGP-like projects that follow the CoP principles are a promising tool for teaching IE, allowing students to build up their 'real-life' experience in a supported, scaffolded space, and to gain knowledge through practice. However, it was not possible to fully simulate a real-life situation.

As one of the first studies to examine the link between a specific teaching method and IE-related learning outcomes, its findings should be taken as preliminary evidence highlighting the need for more research in the area of IE pedagogy. In particular, our research centered on subjective assessments of students' perceptions. While accounting for only a small percentage of the final grade, these could be subject to social desirability bias. Future research should consider evaluation of a broader repertoire of pedagogical tools for teaching IE and using more objective measures of achievement of student outcomes.

Besides adding to the IE literature, our study also contributes to the literature on CoP. Recent studies examined whether virtual spaces are useful for sharing knowledge (Brown et al. 2013) and some suggested that virtual CoP tends to elicit sharing of explicit rather than tacit knowledge (Aljuwaiber 2016). Our study suggests that the GBGP structure coupled with the dedicated online space allowed students to experiment and develop skills that enhanced their confidence within key areas of the IE domain.

**Author Contributions:** Conceptualization, Martina Musteen and Maria Ripollés; Data curation, Andreu Blesa; Formal analysis, Nuno Arroteia; Methodology, Ross Curran.

**Funding:** This work was supported by Universitat Jaume I [Research Group Support Programme. Action 5.2. Continuity Grants for Research Groups].

**Conflicts of Interest:** The authors declare no conflict of interest.

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
