# Peer review of "A Community of Practice Approach to Teaching International Entrepreneurship"

_admsci, doi:10.3390/admsci8040056_

Round 1

Reviewer 1 Report

The paper addresses a very interesting topic, that is, the application of the innovative concept of Communities of Practice (CoP) in the framework of International Entrepreneurship. The literature review is considered adequate and the aims are presented in a straightforward way. The paper has merit to be published, although it does not reveal in a clear way what were the practical implications of the implementation of this CoP. In face of this, I recommend the acceptance of the paper.  

Author Response

A Community of Practice Approach to Teaching International Entrepreneurship

Response to Reviewers

Reviewer 1

1.      Moderate English changes required

Thank you for highlighting this, an additional proof of the document has been carried out prior to resubmission.

2.      The authors have identified an interesting research gap, in the area of pedagogy in IE.

Thank you for your positive comment highlighting the value of the research gap we identify.

3.      I would strongly challenge the possibility of employing the Community of Practice as leading framework for educational and virtual context.  Whereas there are virtual CoP, and please refer to this literature in the paper (see for example work by Dube, Bourhis and Jacob 2005 from J.of Organizational Change Management), for educational context it is hard to talk about naturally emerging community. Communities of Practice in the literature are generally claimed to be organically emerging groups. Given the educational setting, the requirements of combining students from various countries and various institutions, one recognizes huge organizational, motivational and supervisory effort in making students work and making them involved throughout the IE online learning. Please convince the reviewer and the potential readers, that CoP framework is applicable in the setting of your study. This needs to be reflected in the literature section and in methods section of the paper.

Thank you for pointing us to some interesting, relevant literature which we now incorporated in the revised version of the manuscript. We have taken your comments on board and added substantially to our review of the CoP literature. On page3-4, we acknowledge that the initial research on CoP focused on the organic (emergent) aspect of the phenomenon. However, we have also found substantial literature that provides strong evidence that deliberate CoPs can be defined and cultivated explicitly by organizations to achieve specific learning goals. In other words, CoPs can be also applied as a specific methodology used for specific learning purposes, in particular, when the joint practice is supposed to be an important learning element, engendering situated learning (Handley et al. 2006; Pharo et al. 2014). Examples of this instrumental usage of CoP can be found in organizations to foster knowledge sharing among employees, named as organizational communities of practices (see Koliba and Gajda, 2009; Lee and Williams, 2007; or Aljuwaiber 2016). Authors such as Hodge et al. (2014) or Howlett et al. (2016) support the usage of deliberate CoP in an educational context to enhance students’ situated learning. We believe this literature provides a convincing argument that the deliberate CoP approach can be considered in an educational context to offer a space of learning in which students can experiment with different tasks requiring communication and interaction among them. On page 4, we added Table 1 to further clarify the differences between the emergent and deliberate CoP (which we adopt in the GBGP design). Besides the additions to the Literature Review, we have also revised the Methodology section in response to your comment. Specifically, we describe in greater detail how the collaboration between and among student teams was encouraged and facilitated. The key point here is that communication and knowledge sharing in the course of GBGP beyond simple teamwork supporting the deliberate CoP concept. We hope you agree these revisions make this point more convincing.

4.      The authors employ “qualitative analysis of student essays”. What specific methods or techniques have been employed in this qualitative analysis? The reviewer would kindly like to ask the authors to provide comments and explanations in this area. Once the method is put forward, the reviewer will be able to evaluate the quality of the data analysis.

Based on your comment, we have largely overhauled the Methods section. We provide a far more detailed description of the data and the partial grounded approach we used to analyze it.  Specifically, on page 8, we posit that this approach is suitable given that the theory related to IE pedagogy is still underdeveloped. Citing studies that used this approach (e.g., Jack et al. 2008; Sunduramurthy et al. 2016), we describe the individual phases that begin with research questions and some theoretical a priori understanding of the phenomenon while allowing that “the data speak” (Shepherd and Sutcliffe 2011). We detail our findings with respect to the quotes supporting the literature-driven themes as well as new, emerging categories in Table 4.

5.      If the authors decide to use some quotes, it is advisable to make them much shorter.

Thank you for this point. In the revised manuscript, we now present only very short quotes in the text leaving examples of illustrative quotes for each theme for Table 4.

6.      Also, they could reconsider thinking of revising how they see the purpose of this research. The authors claim that they aim to analyse the effectiveness of an experiential teaching innovation. The reviewer is not convinced whether “qualitative analysis” you claim to have employed allows you to evaluate the effectiveness of this method (e.g. line 31, p. 1; line 393, p. 11). This should be revised.

Thank you for highlighting this issue. The study seeks to address a gap in our understanding of the potential contribution of adopting a CoP approach in International Entrepreneurship education. We feel our responses to the definitional, and methodological issues raised in earlier comments also help with this issue. Further, we clarify on page 1:

given the dearth of theory regarding IE pedagogy, we use the partially grounded approach to assess this teaching innovation, designed as a Community of Practice (CoP), through analysis of students’ self-perception of their abilities related to defining, recognizing and evaluating international business opportunities, designing and validating a business model based on such opportunities and creating a plan for pursuing these opportunities. Our study also provides some evidence that, by promoting learning through practice, the CoP based teaching method has impacted upon students’ emotions, self-efficacy and self-perceptions of their entrepreneurial intentions”

We feel these revisions better clarify the purpose of the research.

7.      It would also be very helpful for the paper to make it very clear throughout the whole paper what is meant by Global Board Game.

Thank you for this comment. We have made sure to clarify and more clearly communicate in the Introduction section what the GBGP is. Specifically, we state on page 1:

“The GBGP involves semi-structured online collaboration between undergraduate student teams from three different countries to ideate, develop, and market a product (a board game) to another country. The emphasis is on communication within and between teams to learn, experiment, and test assumptions towards creating a tangible board game prototype and develop a viable market entry plan.”

We feel this addresses the clarity issue you highlighted.

8.      A good idea would be to provide more insights into the progress and stages of student work during the GBG project.

Thank you for highlighting this point. Following the definitional clarifications detailed in the above response, we have also now included a table illustrating the globally aligned milestones of the project and the timeline they were delivered on (see Table 2). We hope this will more clearly convey the different stages of the GBGP. 

Reviewer 2 Report

The authors have identified an interesting research gap, in the area of pedagogy in IE.

I would strongly challenge the possibility of employing the Community of Practice as leading framework  for educational and virtual context.  Whereas there are virtual CoP, and please refer to this literature in the paper (see for example work by Dube, Bourhis and Jacob 2005 from J.of Organizational Change Management),  for educational context it is hard to talk about naturally emerging community. Communities of Practice in the literature are generally claimed to be organically emergin groups. Given the educational setting, the requirements of combining students from various countries and various institutions, one recognizes huge organizational, motivational and supervisory effort in making students work and making them involved throughout the IE online learning. Please convince the reviewer and the potential readers, that CoP framework is applicable in the setting of your study. This needs to be reflected in the literature section and in methods section of the paper.

The authors employ „qualitative analysis of student essays”. What specific methods or techniques have been employed in this qualitative analysis? The reviewer would kindly like to ask the authors to provide comments and explanations in this area. Once the method is put forward, the reviewer will be able to evaluate the qulity of the data analysis.

What is the outcome of this analysis? Is it presented in Figure 1? If that is what the authors put forward, in the reviewer's view, they basically repeat what they have presented in literature review and board game description aims and guidelines.

If the authors decide to use some quotes, it is advisable to make them much shorter.

Also, they could reconsider thinking of revising how they see the purpose of this research. The authors claim that they aim to analyse the effectiveness of an experiential teaching innovation. The reviewer is not convinced  whether “qualitative analysis”  you claim to have employed allows you to evaluate the effectiveness of this method (e.g. line 31, p. 1; line 393, p. 11). This should be revised.

It would also be very helpful for the paper to make it very clear throughout the whole paper what is meant by Global Board Game.

A good idea would be to provide more insights into the progress and stages of student work during the GBG project.

References:

Line Dubé, Anne Bourhis, Réal Jacob,  (2005)  "The impact of structuring characteristics on the launching of virtual communities of practice", Journal of Organizational Change Management, Vol. 18 Issue: 2, pp.145-166, https://doi.org/10.1108/09534810510589570

Author Response

A Community of Practice Approach to Teaching International Entrepreneurship

Response to Reviewers

Reviewer 2

1.      I enjoyed reading your paper entitled A Community of Practice Approach to Teaching International Entrepreneurship.

Thank you, we are delighted you enjoyed the paper, and have carefully reflected upon, and responded to, your constructive review.

2.      The research question(s) of the paper is (are) not clear. You mention in the abstract that “the study evaluates an experiential teaching innovation in the area of international entrepreneurship, the Global Board Game project”. However, in the Introduction section you mention that “Specifically, we evaluate this teaching innovation in terms of students’ self-perception of their abilities related to defining, recognizing and evaluating international business opportunities, designing and validating a business model based on such opportunities and creating a plan for pursuing these opportunities” It seems to me that the subjects assessed more the GBG project by itself and not international business opportunities, for instance.

You are correct and thank you for pointing this out. Indeed, the aim of our study was to evaluate the GBGP as an experiential teaching innovation in the area of international entrepreneurship as is stated in the Abstract. Our research is not actually measuring student’s abilities in terms of grade/exam performance/business start-up, but through qualitatively examining their reflections on their participation in the GBGP. Thus, we have assessed their self-perceptions related to such abilities along with the impact of the GBGP on their attitudes. We clarify this in the Introduction section on page 1 with the following:

“…through analysis of students’ self-perception of their abilities related to defining, recognizing and evaluating international business opportunities, designing and validating a business model based on such opportunities and creating a plan for pursuing these opportunities.”

3.      How you established, for sample subjects (the students), the criteria for inclusion? Based on their interest or was it part of their mandatory curricula? It is important to have a clear image of your study to be able to assess the bias in filling the essays which the students submitted at the end of the semester. As such, an additional issue may emerge since the learning outcomes may be defined differently by each university curricula, and you have students from 3 universities involved. At the same time, you are mentioning that your subjects were students enrolled in international entrepreneurship and marketing courses at SDSU, UJI and AU. While for those enrolled in international entrepreneurship the reason is clear, what was the reason to include marketing students?

Thank you for pointing out this important issue. As we now detail in the revised Data Collection and Analysis section of our paper all students participating in the GBGP were encouraged to submit their reflective summaries on their experiences (see pages 7-8). Furthermore, we now address the bias concern through incorporating the following in the paper:

“It is important to note that in order to motivate the students to submit their summaries, they accounted for a small percentage (around 2%) of the final grade. At the same time, to minimize social desirability bias, we assured the students that they could reflect on both positive and negative aspects of the project and would not be penalized for any negative comments offered. In addition, we emphasized that their insights will be used to improve the project.” (page 7).

In terms of the student learning outcomes (SLOs), the five instructors/researchers engaged in extensive discussions over a period of several months prior to and during the design of the GBGP. This was to ensure that the intended SLOs for all students across all classes (international entrepreneurship and international marketing) were identical. As the instructors of the International Marketing class stated, the GBGP fits well within the curriculum given the strong emphasis on international business development and international entrepreneurship. We hope this better clarifies this aspect of the research.

4.      The main weakness of the paper is related to students’ attitude toward entrepreneurial intention. Even though there is a body of literature linking prior knowledge to entrepreneurial intention, it is not the type of knowledge you are assuming but more about the business or prior experiences of the entrepreneur.

So, a controlled environment like GBG enabled by Ideator is more of a lab experiment than a real life situation assessment.

We agree with you that our findings related to entrepreneurial intention must be viewed and interpreted with caution. While we find evidence that a number of students (reflected in 33 quotes see Table 4) have expressed that participation in the GBGP made them consider or even desire pursuing entrepreneurial careers, we cannot claim that these students will indeed become entrepreneurs or that their intention will persist. In the revised version of our manuscript we discuss this weakness in the Discussion section (page 17). We also acknowledge that the aim of the GBGP was to allow students to build up their ‘real life’ experience in a supported, scaffolded space. However, it was not possible to fully simulate a real life situation.

5.      In terms of terms definition, I am more sympathetic to a narrower definition of CoP. You are embracing Wenger approach, while in my opinion Brown & Duguid or Botha definitions, emphasizing expertise as a critical element of any CoP members, are more appropriate. So, students are lacking expertise by definition, so they cannot create real CoP but rather teams.

We agree with you that initial research on CoP has viewed the phenomenon primarily as emergent/organic, requiring significant expertise of the CoP members. Your comment made us go back to the literature see if we can provide a more convincing argument that a deliberate form of CoP can be also used successfully in the education setting. For example, recent literature (Hodge et al. 2014; Howlett et al. 2016) supports the usage of deliberate CoP in an educational context to enhance students’ situated learning. We adopt this approach while further clarifying in the revised Literature Review on page 7, and in Table 1. Furthermore, the Methods section (page 7) maps the GBGP against the characteristics of a deliberate CoP (See Table 3) to clarify how this approach was incorporated.

6.      The conclusions are trivial, not supported by results and represent rather general statements. For instance, you mention that “The qualitative analysis of student reflections on the project provided evidence that the GBG project, as a teaching tool, was effective in helping students create a more tangible link between IE theory and practice. In addition, it indicated that participating in the project influenced students ‘attitudes toward entrepreneurship as a career path”. I fail to see any evidence for that in your paper. Hence, I cannot agree with your conclusions, but not necessarily because they are wrong but rather because they are not substantiated. You do not provide data about your research, I am guessing on what your results actually are. Not even an Appendix with results from the software tool you used are not provided.

We appreciate your well-made point and have taken on board your suggestion to be more detailed in presenting the approach to the data analysis as well as findings. In response to your comment, we have overhauled both the Methods and Findings section and revised the Conclusions as well. Specifically, we detail the phases of the “partially grounded” approach that we use to analyze our data and present a Table with the number of quotes that support both a priori as well as emergent themes arising from the data (See Table 4). While we see our findings as only a preliminary evidence and acknowledge some of the weaknesses associated with our approach, we feel our findings present some interesting insights and should be of interest to IE educators seeking to either adopt the GBGP or to develop a similar teaching method in their classes.  We hope you agree.

7.      Overall, your paper lacks an in-depth analysis. It lacks detail in many of its sections: literature review (even thought it was not mentioned as such), results, discussion conclusion and.

The references are not appropriately inserted in the text and your paper is not observing the journal requirements.

We feel our responses to the previous issues contribute to the aspects of the manuscript mentioned here. All sections have been subject to extensive revision and additional detail relating to the analytical approach used, and the limitations of the study offered.

Thank you for raising this issue, the manuscript has been further reviewed to conform to the journal guidelines.

Reviewer 3 Report

Dear Author(s),

I enjoyed reading your paper entitled A Community of Practice Approach to Teaching 3 International Entrepreneurship. However, in my opinion your paper is not ready yet to be published in its current form.

These are my comments and suggestion to further improve your paper:

1. The research question(s) of the paper is (are) not clear. You mention in the abstract that “the study evaluates an experiential teaching innovation in the area of international entrepreneurship, the Global Board Game project”.

However, in the Introduction section you mention that “Specifically, we evaluate this teaching innovation in terms of students’ self-perception of their abilities related to defining, recognizing and evaluating international business opportunities, designing and validating a business model based on such opportunities and creating a plan for pursuing these opportunities”.

It seems to me that the subjects assessed more the GBG project by itself and not international business opportunities, for instance.

2. How you established, for sample subjects (the students), the criteria for inclusion? Based on their interest or was it part of their mandatory curricula? It is important to have a clear image of your study to be able to assess the bias in filling the essays which the students submitted at the end of the semester. As such, an additional issue may emerge since the learning outcomes may be defined differently by each university curricula, and you have students from 3 universities involved. At the same time, you are mentioning that your subjects were students enrolled in international entrepreneurship and marketing courses at SDSU, UJI and AU. While for those enrolled in international entrepreneurship the reason is clear, what was the reason to include marketing students?

3. The main weakness of the paper is related to students’ attitude toward entrepreneurial intention. Even though there is a body of literature linking prior knowledge to entrepreneurial intention, it is not the type of knowledge you are assuming but more about the business or prior experiences of the entrepreneur. So, a controlled environment like GBG enabled by Ideator is more of a lab experiment than a real life situation assessment.

4. In terms of terms definition, I am more sympathetic to a narrower definition of CoP. You are embracing Wenger approach, while in my opinion Brown & Duguid or Botha definitions, emphasizing expertise as a critical element of any CoP members, are more appropriate. So, students are lacking expertise by definition, so they cannot create real CoP but rather teams.

5. The conclusions are trivial, not supported by results and represent rather general statements. For instance, you mention that “The qualitative analysis of student reflections on the project provided evidence that the GBG project, as a teaching tool, was effective in helping students create a more tangible link between IE theory and practice. In addition, it indicated that participating in the project influenced students ‘attitudes toward entrepreneurship as a career path”. I fail to see any evidence for that in your paper. Hence, I cannot agree with your conclusions, but not necessarily because they are wrong but rather because they are not substantiated. You do not provide data about your research, I am guessing on what your results actually are. Not even an Appendix with results from the software tool you used are not provided.

6. Overall, your paper lacks an in-depth analysis. It lacks detail in many of its sections: literature review (even thought it was not mentioned as such), results, discussion conclusion and. The references are not appropriately inserted in the text and you paper is not observing the journal requirements.

Author Response

A Community of Practice Approach to Teaching International Entrepreneurship

Response to Reviewers

Reviewer 3

1.      English language and style are fine/minor spell check required.

Thank you for highlighting this, a further check of the manuscript has been conducted.

2.      …the authors offer a relatively innovative project of international education.

Thank for your kind comments.

3.      The author (s) provide a wide array of studies mostly related to the field of international entrepreneurship, and this literature provides an landscape of theoretical microfoundations for the education model here described.

Thank you for your comment acknowledging the contribution this study makes.

4.      In my opinion, the study present contributes for this research area.

Thank you for your comment highlighting the contribution of this study.

Reviewer 4 Report

The authors could improve the Conclusion section. More contributions of the study  can be inserted.

Abour this article titled "A  Community of Practice Approach to Teaching International Entrepreneurship ", the authors offer a relatively innovative project of international education. The  author (s) provide a wide array of studies mostly related to the field  of international entrepreneurship, and this literature provides an  landscape of theoretical  microfoundations for the education model here described. In my opinio,  the study present contributes for this research area.

Author Response

We appreciate your well-made point and have taken on board your suggestion. In response to your comment, we have revised the Conclusions. While we see our findings as only a preliminary evidence and acknowledge some of the weaknesses associated with our approach, we feel our findings present some interesting insights and should be of interest to IE educators seeking to either adopt the GBGP or to develop a similar teaching method in their classes.  We hope you agree.

Round 2

Reviewer 2 Report

The author(s) employ(s) unclear methodological approach.

The authors employ „qualitative analysis of student essays”. What specific methods or techniques have been employed in this qualitative analysis? They claim to emply "partially grounded" inductive approach but they do not clarify the "partiality" dimension. What is different about it, what is this approach?

Also, the main constructs that emerged from the research are not based on insightful qualitative analysis but on simple numbers of quotes behind the related constrcuts and dimensions. How have the main constructs emerged?

It is still unclear what sampling approach the author(s) has(have) employed.

Author Response

We would like to thank the reviewers for their additional constructive input into the manuscript. We have carefully considered0 these and offer the following responses which highlight how the comments have stimulated our revision of the manuscript.

Reviewer 2, Round 2:

Reviewer 2: The authors employ unclear methodological approach. The authors employ “qualitative analysis of student essays”. What specific methods or techniques have been employed in this qualitative analysis? They claim to employ “partially grounded” inductive approach but they do no clarify the “partiality” dimension. What is different about it, what is this approach?

Response: We apologize for the lack of clarity regarding our analytical approach and hope that the detail we added to the revised Methodology section (pg. 5 –13) will address your remaining concerns. Specifically, we put additional effort into elucidating the partially grounded approach which we used to qualitatively analyze the data in our study. The partially grounded approach differs from the purely grounded approach to data analysis in several respects (Ahsan et al, 2018; Shepherd & Sutcliffe, 2011). The purely grounded approach is used to derive new theory solely from data (Glaser and Strauss, 1967).  Researchers pursuing the purely grounded, inductive approach often take a position of “unknowing” (Shepherd and Sutcliffe, 2011, pg. 361), approaching the raw data without preconceived notions about the meaning they are searching for. The partially grounded approach is also inductive in that data is used to better understand a phenomenon and derive concepts and constructs. However, the difference is that, in the partially grounded approach, the analysis is guided by existing literature. In other words, the partially grounded approach connects and compares the meaning emerging from the raw data with existing concepts and constructs (Ahsan et al., 2018).  In that sense, the “partiality” comes from the fact that researchers come to the data influenced by their familiarity with the relevant literature. 

In our study, adopting the partially grounded approach, we engaged in extensive analysis of the data contained in the reflective summaries of students participating in GBGP while being informed by the previous entrepreneurship education (EE), IE and CoP literature. Thus, our coding included search for language indicative of students’ learning related to the following domains that constituted our “preconceived notions” regarding the data at hand:

(1) defining, recognizing and evaluating international business opportunities,

(2) designing and validating a business model for such an opportunity

(3) creating an offering for and translating the proposed business model to a specific international market.

(4) learning through knowledge sharing and practice

(5) entrepreneurial intention

The above themes therefore constitute the “partiality” in our approach. To be true to the “grounded” aspect of our analysis, however, we have also been open to additional themes that emerge from our data, themes that have not been determined a priori.  These include cross-cultural competences, emotions, and challenges and are elaborated in the Study Findings section.

Again, we hope that this additional information and references to studies that have adopted the same approach addresses your concern.

Also, the main constructs that emerged from the research are not based on insightful qualitative analysis but on simple numbers of quotes behind the related constructs and dimensions. How have the main constructs emerged?

Response: We are sorry if we gave you the impression that the key constructs were based on a simple number of quotes. We provide the number of quotes as an illustration of the prevalence of the language indicative of each theme. As we now detail in the revised manuscript (pg.8- 9), the critical component in the process of identifying the constructs was the open and axiel coding, techniques adopted from Strauss and Corbin (1988). Specifically, during the open coding phase, three researchers independently coded the textual data categorizing specific quotes into 28 different categories maintained in RQDA, a software package that is widely used by qualitative researchers. These concepts were either in vivo, occurring frequently in the analyzed texts, or were given labels based on the researchers’ interpretations of emerging concepts (Sundaramurthy et al., 2013) or a connection to a priori constructs. The axiel coding phase involved the research team revising these codes by comparing, discussing and debating the themes and concepts that emerged during independent analysis. This process also involved giving meaning to data that were fractured during the open coding phase (Strauss and Corbin, 1988). The goal was to determine whether students participating in GBGP indicated learning in the five key domains in IE and CoP (specified a priori based on our literature review). The newly emerged themes also indicated that GBGP participation had impact on student’s emotions, cross-cultural competences and posed challenges in terms of primarily communication-related dynamics.

It is important to note that the insights gained through our careful and intensive qualitative analysis should by no means be interpreted as an evidence confirming or disconfirming previous theory. As is the case with studies following the partially grounded approach (e.g. Ahsan et al., 2018; Jack et al., 2008; Sundaramurthy et al, 2013; 2016), ours seeks to provide greater understanding of a phenomenon that is still poorly understood. Specifically, by identifying experiences, learning and processes associated with GBGP, we hope to contribute insights into the vastly under-researched area of IE pedagogy. While preliminary, these can be helpful to both IE scholars and educators seeking to devise meaningful pedagogical tools.

It is still unclear what sampling approach the author(s) has(have) employed.

As we detail in the revised manuscript, we collected and examined all reflective summaries that were submitted by the students participating in GBGP. That is 86 in total (30 from SDSU, 35 from UJI and 21 from AU). 23 students did not provide any summaries and were therefore not considered in the study. We include this additional detail in the revised manuscript and hope this clarifies the sampling issue.

Thank you for your constructive comments. We believe our manuscript improved substantially as a result of your thoughtful review.

Reviewer 3 Report

You improved the paper. I may still find less satisfactory the fact that you still does not consider all my comments.

Author Response

Reviewer 3, Round 2.

You improved the paper. I may still find less satisfactory the fact that you still does not consider all my comments.

Response: Thank you for your positive comments regarding our improvement of the paper. Please accept our apologies, we have responded to each of your previous comments at length. However, our response to your review was mislabeled in error. We would like to highlight below how we acted upon your constructive comments, and again apologize for not making this clearer in the previous response. Indeed, it is thanks to the constructive comments you provided that we have been able to improve the paper further as you say. For clarity, we have included the responses we formulated to your points in the last round of review below with the now correct reviewer label. 

Reviewer 3, Round 1.

Reviewer: Point 1.

The research question(s) of the paper is (are) not clear. You mention in the abstract that “the study evaluates an experiential teaching innovation in the area of international entrepreneurship, the Global Board Game project”.

However, in the Introduction section you mention that “Specifically, we evaluate this teaching innovation in terms of students’ self-perception of their abilities related to defining, recognizing and evaluating international business opportunities, designing and validating a business model based on such opportunities and creating a plan for pursuing these opportunities”.

It seems to me that the subjects assessed more the GBG project by itself and not international business opportunities, for instance.

Response: You are correct and thank you for pointing this out. Indeed, the aim of our study was to evaluate the GBGP as an experiential teaching innovation in the area of international entrepreneurship as is stated in the Abstract. Our research is not actually measuring student’s abilities in terms of grade/exam performance/business start-up, but through qualitatively examining their reflections on their participation in the GBGP. Thus, we have assessed their self-perceptions related to such abilities along with the impact of the GBGP on their attitudes. We clarify this in the Introduction section on page 1 with the following:

“…through analysis of students’ self-perception of their abilities related to defining, recognizing and evaluating international business opportunities, designing and validating a business model based on such opportunities and creating a plan for pursuing these opportunities.”

Reviewer: Point 2.

How you established, for sample subjects (the students), the criteria for inclusion? Based on their interest or was it part of their mandatory curricula? It is important to have a clear image of your study to be able to assess the bias in filling the essays which the students submitted at the end of the semester. As such, an additional issue may emerge since the learning outcomes may be defined differently by each university curricula, and you have students from 3 universities involved. At the same time, you are mentioning that your subjects were students enrolled in international entrepreneurship and marketing courses at SDSU, UJI and AU. While for those enrolled in international entrepreneurship the reason is clear, what was the reason to include marketing students?

Response: Thank you for pointing out this important issue. As we now detail in the revised Data Collection and Analysis section of our paper all students participating in the GBGP were encouraged to submit their reflective summaries on their experiences (see pages 7-8). Furthermore, we now address the bias concern through incorporating the following in the paper:

“It is important to note that in order to motivate the students to submit their summaries, they accounted for a small percentage (around 2%) of the final grade. At the same time, to minimize social desirability bias, we assured the students that they could reflect on both positive and negative aspects of the project and would not be penalized for any negative comments offered. In addition, we emphasized that their insights will be used to improve the project.” (page 7).

In terms of the student learning outcomes (SLOs), the five instructors/researchers engaged in extensive discussions over a period of several months prior to and during the design of the GBGP. This was to ensure that the intended SLOs for all students across all classes (international entrepreneurship and international marketing) were identical. As the instructors of the International Marketing class stated, the GBGP fits well within the curriculum given the strong emphasis on international business development and international entrepreneurship. We hope this better clarifies this aspect of the research.

Reviewer: Point 3.

The main weakness of the paper is related to students’ attitude toward entrepreneurial intention. Even though there is a body of literature linking prior knowledge to entrepreneurial intention, it is not the type of knowledge you are assuming but more about the business or prior experiences of the entrepreneur.

So, a controlled environment like GBG enabled by Ideator is more of a lab experiment than a real life situation assessment.

Response: We agree with you that our findings related to entrepreneurial intention must be viewed and interpreted with caution. While we find evidence that a number of students (reflected in 33 quotes see Table 4) have expressed that participation in the GBGP made them consider or even desire pursuing entrepreneurial careers, we cannot claim that these students will indeed become entrepreneurs or that their intention will persist. In the revised version of our manuscript we discuss this weakness in the Discussion section (page 17). We also acknowledge that the aim of the GBGP was to allow students to build up their ‘real life’ experience in a supported, scaffolded space. However, it was not possible to fully simulate a real life situation.

Reviewer: Point 4.

In terms of terms definition, I am more sympathetic to a narrower definition of CoP. You are embracing Wenger approach, while in my opinion Brown & Duguid or Botha definitions, emphasizing expertise as a critical element of any CoP members, are more appropriate. So, students are lacking expertise by definition, so they cannot create real CoP but rather teams.

Response: We agree with you that initial research on CoP has viewed the phenomenon primarily as emergent/organic, requiring significant expertise of the CoP members. Your comment made us go back to the literature see if we can provide a more convincing argument that a deliberate form of CoP can be also used successfully in the education setting. For example, recent literature (Hodge et al. 2014; Howlett et al. 2016) supports the usage of deliberate CoP in an educational context to enhance students’ situated learning. We adopt this approach while further clarifying in the revised Literature Review on page 7, and in Table 1. Furthermore, the Methods section (page 7) maps the GBGP against the characteristics of a deliberate CoP (See Table 3) to clarify how this approach was incorporated.

Reviewer: Point 5.

The conclusions are trivial, not supported by results and represent rather general statements. For instance, you mention that “The qualitative analysis of student reflections on the project provided evidence that the GBG project, as a teaching tool, was effective in helping students create a more tangible link between IE theory and practice. In addition, it indicated that participating in the project influenced students ‘attitudes toward entrepreneurship as a career path”. I fail to see any evidence for that in your paper. Hence, I cannot agree with your conclusions, but not necessarily because they are wrong but rather because they are not substantiated. You do not provide data about your research, I am guessing on what your results actually are. Not even an Appendix with results from the software tool you used are not provided.

Response:  We appreciate your well-made point and have taken on board your suggestion to be more detailed in presenting the approach to the data analysis as well as findings. In response to your comment, we have overhauled both the Methods and Findings section and revised the Conclusions as well. Specifically, we detail the phases of the “partially grounded” approach that we use to analyze our data and present a Table with the number of quotes that support both a priori as well as emergent themes arising from the data (See Table 4). While we see our findings as only a preliminary evidence and acknowledge some of the weaknesses associated with our approach, we feel our findings present some interesting insights and should be of interest to IE educators seeking to either adopt the GBGP or to develop a similar teaching method in their classes.  We hope you agree.

Reviewer: Point 6.

Overall, your paper lacks an in-depth analysis. It lacks detail in many of its sections: literature review (even thought it was not mentioned as such), results, discussion and conclusion. The references are not appropriately inserted in the text and your paper is not observing the journal requirements.

We feel our responses to the previous issues contribute to the aspects of the manuscript mentioned here. All sections have been subject to extensive revision and additional detail relating to the analytical approach used, and the limitations of the study offered.

Thank you for raising this issue, the manuscript has been further reviewed to conform to the journal guidelines.

Additional References

Ahsan, M., Zheng, C., DeNoble, A., & Musteen, M. (2018). From Student to Entrepreneur: How Mentorships and Affect Influence Student Venture Launch. Journal Of Small Business Management, 56(1), 76-102.

Sundaramurthy, C., Musteen, M., & Randel, A. 2013. Social value creation: A qualitative study of social entrepreneurs in India. Journal of Developmental Entrepreneurship, 18: 1-21.

Strauss, A and J Corbin (1998). Basics of Qualitative Research: Techniques and Procedures for Developing Grounded Theory. Thousand Oaks, CA: Sage Publications.

Round 3

Reviewer 2 Report

Dear Authors,

Thank you for improving the paper. I would like to congratulate you on your effort.

I am sure that the readers will find the paper of much interest.

Please correct the axiel to axial coding spelling and few other typos.

Congratulations!  Feel encouraged to submit other papers of academic value to the journal

Reviewer

Author Response

The authors would like to thank the reviewers for their positive and helpful comments, and for the very kind comments received regarding our improvement of the paper. We offer our responses to your comments below.

Thank you for improving the paper. I would like to congratulate you on your effort.

I am sure that the readers will find the paper of much interest.

Response: Thank you for your kind words, the improvement to the paper is thanks to the thoughtful contributions of the reviewers.

Please correct the axiel to axial coding spelling and few other typos.

Congratulations!  Feel encouraged to submit other papers of academic value to the journal

Response: Thank you, this has now been corrected and the paper has been further proofed. 

Reviewer 3 Report

Overall, you provide answers to my main points of concern. Your Literature review may benefit from a wide array of papers, for instance Ceptureanu S. I., Ceptureanu E.G. Role of Knowledge Based Communities in Knowledge Process, Economia Seria Management http://www.management.ase.ro/reveconomia/2015-2/5.pdf may be useful in providing a clear framework for communities of practice and their role in knowledge. Your paper does not comply with the citation style of the journal. It was one of my observations which was not addressed in the revised version. The same for References section.

Author Response

The authors would like to thank the reviewers for their positive and helpful comments, and for the very kind comments received regarding our improvement of the paper. We offer our responses to your comments below. 

Reviewer 3: Overall, you provide answers to my main points of concern.

Response: Thank you for your constructive comments which helped us to enhance the paper.

Reviewer 3: Your Literature review may benefit from a wide array of papers, for instance Ceptureanu S. I., Ceptureanu E.G. Role of Knowledge Based Communities in Knowledge Process, Economia Seria Management http://www.management.ase.ro/reveconomia/2015-2/5.pdf may be useful in providing a clear framework for communities of practice and their role in knowledge.

Response: Thank you for highlighting this interesting article. This paper has now been incorporated to strengthen our description of CoP, in particular the usefulness and prevalence of CoP in organizational settings.  

Reviewer 3: Your paper does not comply with the citation style of the journal. It was one of my observations which was not addressed in the revised version. The same for References section.

Response: Thank you for highlighting this, we have re-checked the citations and reference list to ensure we conform to the referencing guidelines as per the Reference List and Citations Style Guide. We have corrected some errors in the referencing following your suggestion